# Association between dual smoking and dyslipidemia in South Korean adults

**Wonseok Jeong**[ID]*

Department of Public Health, Graduate School, Seoul National University, Seoul, Republic of Korea

* wsjeong22@snu.ac.kr

**Editor:** Negar Rezaei, Non-Communicable Diseases Research Center, Endocrinology and Metabolism Population Sciences Institute, Tehran University of Medical Sciences, ISLAMIC REPUBLIC OF IRAN

**Data Availability Statement:** All Knhanes files are available from the Knhanes database (https://knhanes.kdca.go.kr/knhanes/sub03/sub03_02_05.do).

## Abstract

### Objectives

Dyslipidemia increases the risk of serious cardiovascular disease; and conventional cigarette smoking is widely recognized as a risk factor. Thus, as electronic cigarettes were introduced, many smokers utilized them for smoking cessation. However, due to the lack of researches scrutinized the relationship between dual smoking and dyslipidemia, a lot of those who failed in cessation end up utilizing both types of cigarettes (dual smoking). Therefore, this study aimed to examine the effect of dual smoking on dyslipidemia in South Korean adults.

### Methods

Data were included from the 2013, 2014, 2015, 2016, and 2019 Korean National Health and Nutrition Examination Survey of 8,398 Korean men. The KNHANES is conducted by a national institution based on random cluster sampling, and therefore, the data gained from it is statistically reliable and representative in comparison to surveys performed by private institutions. Cigarette use status was the main independent variable. Cigarette use status was categorized as one of the four status: dual smoker, single smoker, non-smoker, and ex-smoker. The dependent variable, presence of dyslipidemia, was defined according to the National Cholesterol Education Program as displaying one or more of the following criteria: total cholesterol: ≥240 mg/dL, LDL cholesterol: ≥160 mg/dL, triglyceride: ≥200 mg/dL, or HDL cholesterol: ≤40 mg/dL. Multiple logistic regression analysis was performed to examine target association.

### Results

Current and former users of either electronic or conventional cigarettes presented with a higher odd ratio of dyslipidemia compared to non-smokers. (ex-smoker: OR = 1.60, 95% CI = 1.41–1.83; single smoker: OR = 1.21, 95% CI = 1.06–1.38). More importantly, those who smoke both conventional and electronic cigarettes were also, at high risk of dyslipidemia (dual smoker: OR = 1.66, 95% CI = 1.26–2.20). Along with smoking status, those who did not exercise had a higher risk of dyslipidemia than those who exercised regularly and higher self-reported health status was also related to a lower risk of dyslipidemia.

**Funding:** The authors received no specific funding for this work.

**Competing interests:** The authors have declared that no competing interests exist.

## Conclusion

This study suggests that along with conventional cigarettes, dual smoking negatively impacts dyslipidemia, and smoking cessation of evert types is necessary for a healthy life.

## 1. Introduction

Dyslipidemia refers to an abnormal level of lipids in the blood; and is diagnosed based on the exhibition one or more of the following criteria: increased levels of total cholesterol, low-density lipoprotein (LDL) cholesterol, or triglycerides; or a decreased level of high-density lipoprotein (HDL) cholesterol [1]. Dyslipidemia progresses slowly without distinctive symptoms, making it difficult for individuals to recognize the early stages. As it is often asymptomatic, it is usually left untreated, and individuals with dyslipidemia frequently experience severe health outcomes later. Correspondingly, in South Korea, among well-established risk factors for cardiovascular disease, the age standardized prevalence was the highest for dyslipidemia, while awareness was the lowest [2]. Owing to the lack of obvious symptoms, prevention, through the investigation of factors associated with dyslipidemia, is critical.

Conventional cigarette smoking is also widely recognized as a risk factor for the occurrence of cerebrovascular, cardiovascular, and other circulatory diseases [3]. Historically, conventional cigarette smoking was popular in South Korea. In the early 1980s, it was estimated that 8 out of 10 Korean men smoked cigarettes [4, 5]. However, as extensive research has reported the serious long-term effects of smoking conventional cigarettes, general awareness to quit smoking has increased dramatically. Hence, Korean smokers have undertaken various strategies to quit smoking. Since the introduction of electronic cigarettes in the country, many smokers have attempted to utilize them for smoking cessation; according to ITC surveys conducted in 2010, the electronic cigarettes trial use and current use of e-cigarettes were 11% and 7%, respectively, among smokers in Korea [6]. Yet, because of the high addictiveness of cigarettes, few individuals succeed in switching from conventional to electronic cigarettes, and many end up utilizing both types of cigarettes [7]. People who smoke both electronic and conventional cigarettes are referred as dual smokers.

Conventional cigarette smoking is a leading contributing factor to dyslipidemia, and quantitative studies have reported on the relationship between the two [3, 8]. This has led to major changes in the smoking behavior of Korean men [8]. Furthermore, quantitative studies have shown the adverse health effects of electronic cigarettes (e-cigarettes) [9]. Despite the fact that most smoking cessation attempts using e-cigarettes result in dual smoking, limited research has scrutinized the relationship between dual smoking and dyslipidemia [10]. This gap in the research led us to investigate the potential connection between dyslipidemia and smoking behaviors, including dual smoking as a key indicator.

The objective of this study was to examine the association between dual smoking and dyslipidemia in Korean men, comparing those with and without dyslipidemia among dual-, single-, ex-, and non-smokers.

## 2. Materials and methods

### Data and study participants

Data for this study were taken from a sample of the 2013, 2014, 2015, 2016, and 2019 Korean National Health and Nutrition Examination Survey (KNHANES), conducted by the Korea

Centers for Disease Control and Prevention, which collects information on the health of the public, the status of chronic diseases, and smoking/drinking behaviors. From 2018, the ethics approval for the KNHANES was waived by the KCDC Institutional Review Board under the Bioethics & Safety Act and opened to the public. The study protocol has been priorly approved by the Institution's ethics committee on research on humans. The board waived the need for informed consent, as the subjects' records and information were anonymized and de-identified prior to analysis [11]. Also, all participants provided informed consent to participate in the KYRBWS and were guaranteed anonymity. The KNHANES offers a complex, stratified, multistage, probability-cluster survey with rolling sampling designs to analyze a representative, civilian, noninstitutionalized South Korean population. Along with stratification of the geographic areas (16 provinces of South Korea), probability-clustered sampling methods were performed in two steps. First, the primary sampling units were formed by sex, 26 age groups, and 24 land and housing classes. Second, 20 families were randomly sampled in each primary sampling unit. Then, each member of the sampled family provided written informed consent. Thus, the data from KNHANES can be considered to represent the entire population of South Korea [12].

Of the 39,208 individuals who participated in the surveys, I first excluded those aged <19 years (n = 31,022); those under 19 years were excluded since the minimum age for smoking in South Korea is 20 years. Among 31,022, 17,002 remained due to the missing covariates such as heavy drinking status, parental inheritance of dyslipidemia, and exercise sessions performed in a week. Lastly, females were excluded, and the final study sample size was 8,398. Korean women were excluded due to possible reporting bias [13] (Fig 1).

## Variables

The primary independent variable was cigarette use status. Participants were classified into four groups: dual smoker, single smoker, ex-smoker, and non-smoker. Individuals who answered "No" to the questions "Have you ever smoked a conventional cigarette?" and "Have you ever vaped an electronic cigarette?" were placed in the "non-smoker" group. Those who reported "Yes" to either of the questions were asked the following follow-up question: "In the past 30 days, have you smoked/vaped a conventional/ electronic cigarettes?" Based on their answers, individuals were placed into the "dual smoker," "single smoker," or the "ex-smoker" groups. Additionally, all analyses included participants' demographic, socioeconomic, and health-related characteristics.

The demographic characteristics included in the study were participants' age (19−39, 40−49, 50−59, ≥60), gender, and marital status (married, single and widowed, separated, or divorced). Socioeconomic factors included participants' education (middle school or lower, high school, college or higher), region (urban or rural), household income (low, medium-low, medium-high, and high), and occupation. Occupation category followed the Korean version of the Standard Classification of Occupations, which were reclassified into four categories: white (office work), pink (sales and service), blue (forestry, fishery, armed forces occupation, and agriculture), and unemployed [14].

The health-related characteristics included the participants' frequency of heavy drinking, defined as more than seven glasses of alcohol, (less than once a month, once a week, and almost every day), the number of days in a week that exercise was performed (none, 1−2 days, 3−4 days, 5−6 days, and every day), self-reported health status (good, normal, and bad), stress level (high, middle, and low), parental history of dyslipidemia (yes, no), and weight changes. The weight changes category directly asked the participants whether their current body weight has increased, decreased, or remained constant over the last year.

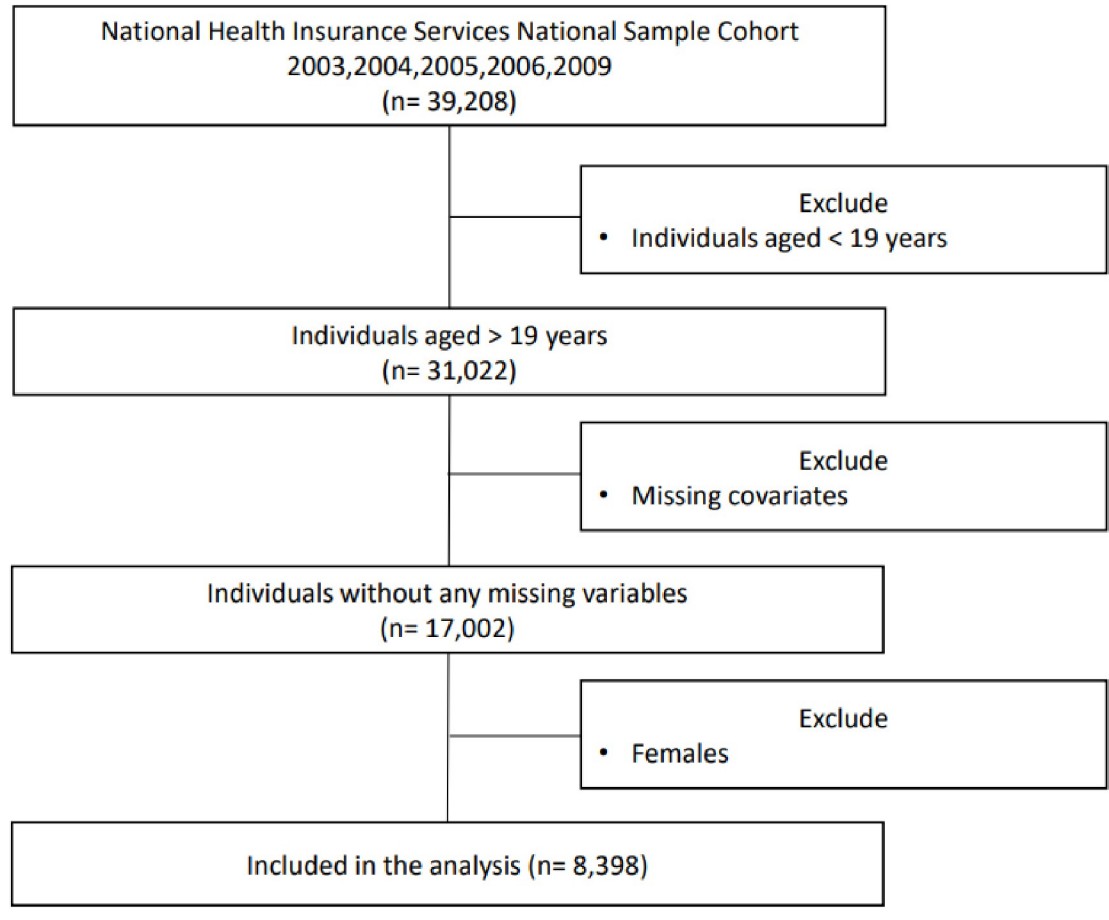

**Fig 1. Flowchart of the participant selection.**

Dyslipidemia was included as the main dependent variable in this study. Dyslipidemia was defined according to the National Cholesterol Education Program, as displaying one or more of the following criteria: total cholesterol: ≥240 mg/dL, LDL cholesterol: ≥160 mg/dL, triglyceride: ≥200 mg/dL, or HDL cholesterol: ≤40 mg/dL [15].

## Statistical analysis

Chi-square tests were conducted to analyze the general characteristics of the study population. A multiple logistic regression analysis was performed to examine the associations between dyslipidemia and smoking behaviors after accounting for potential confounding variables, including demographic, socioeconomic, and health-related characteristics. Results are reported as odds ratios (OR) with a 95% confidence interval (CI). Differences were considered statistically significant with a $p$-value of <0.05. All data analyses were conducted using SAS 9.4 software (version 9.4; SAS Institute Inc., Cary, NC, USA).

## 3. Results

A total of 39,208 participants were included. After all exclusions, data from 8,398 participants were analyzed. Table 1 presents the general characteristics of the study population. According to their total cholesterol, LDL cholesterol, triglycerides, and HDL cholesterol levels, 3,384 of 8,398 participants (40.3%) presented with dyslipidemia and 5,014 (59.7%) did not. Dual

**Table 1. General characteristics of the study population (n = 8398).**

| Variables | Total | | Dyslipidemia | | | | P-value |
|---|---|---|---|---|---|---|---|
| | | | Yes | | No | | |
| Total | 8,398 | (100.0) | 3,384 | (40.3) | 5,014 | (59.7) | |
| **Smoking Behavior** | | | | | | | <0.0001 |
| Dual Smoker | 249 | (3.0) | 110 | (44.2) | 139 | (55.8) | |
| "Single" Smoker | 3,106 | (37.0) | 1,424 | (45.8) | 1,682 | (54.2) | |
| Ex-Smoker | 3,228 | (38.4) | 1,289 | (39.9) | 1,939 | (60.1) | |
| Non-Smoker | 1,815 | (21.6) | 561 | (30.9) | 1,254 | (69.1) | |
| **Age (years)** | | | | | | | <0.0001 |
| 20−39 | 2,940 | (35.0) | 958 | (32.6) | 1,982 | (67.4) | |
| 40−49 | 1,718 | (20.5) | 836 | (48.7) | 882 | (51.3) | |
| 50−59 | 1,556 | (18.5) | 739 | (47.5) | 817 | (52.5) | |
| ≥ 60 | 2,184 | (26.0) | 851 | (39.0) | 1,333 | (61.0) | |
| **Educational level** | | | | | | | <0.0001 |
| Middle school or lower | 1,442 | (17.2) | 628 | (43.6) | 814 | (56.4) | |
| High school | 2,397 | (28.5) | 1,041 | (43.4) | 1,356 | (56.6) | |
| College or higher | 4,559 | (54.3) | 1,715 | (37.6) | 2,844 | (62.4) | |
| **Marital status** | | | | | | | <0.0001 |
| Married | 6,050 | (72.0) | 2,621 | (43.3) | 3,429 | (56.7) | |
| Separated or divorced | 430 | (5.1) | 191 | (44.4) | 239 | (55.6) | |
| Unmarried | 1,918 | (22.8) | 572 | (29.8) | 1,346 | (70.2) | |
| **Region** | | | | | | | 0.5096 |
| Urban area | 3,774 | (44.9) | 1,506 | (39.9) | 2,268 | (60.1) | |
| Rural area | 4,624 | (55.1) | 1,878 | (40.6) | 2,746 | (59.4) | |
| **Household income level** | | | | | | | 0.788 |
| Low | 1,042 | (12.4) | 432 | (41.5) | 610 | (58.5) | |
| Lower middle | 2,025 | (24.1) | 802 | (39.6) | 1,223 | (60.4) | |
| Upper middle | 2,466 | (29.4) | 990 | (40.1) | 1,476 | (59.9) | |
| High | 2,865 | (34.1) | 1,160 | (40.5) | 1,705 | (59.5) | |
| **Drinking Status** | | | | | | | 0.034 |
| Heavy drinking | 5,129 | (61.1) | 2,000 | (39.0) | 3,129 | (61.0) | |
| Moderate drinking | 2,388 | (28.4) | 993 | (41.6) | 1,395 | (58.4) | |
| Light drinking | 881 | (10.5) | 391 | (44.4) | 490 | (55.6) | |
| **Occupational classification** | | | | | | | 0.0001 |
| White-collar | 2,676 | (31.9) | 1,125 | (42.0) | 1,551 | (58.0) | |
| Blue-collar | 2,826 | (33.7) | 1,174 | (41.5) | 1,652 | (58.5) | |
| Pink-collar | 992 | (11.8) | 401 | (40.4) | 591 | (59.6) | |
| None | 1,904 | (22.7) | 684 | (35.9) | 1,220 | (64.1) | |
| **Self-Reported Health Status** | | | | | | | <0.0001 |
| High | 3,087 | (36.8) | 1,076 | (34.9) | 2,011 | (65.1) | |
| Middle | 4,257 | (50.7) | 1,809 | (42.5) | 2,448 | (57.5) | |
| Low | 1,054 | (12.6) | 499 | (47.3) | 555 | (52.7) | |
| **Stress Level** | | | | | | | 0.2022 |
| High | 2,062 | (24.6) | 909 | (44.1) | 1,153 | (55.9) | |
| Middle | 5,017 | (59.7) | 1,983 | (39.5) | 3,034 | (60.5) | |
| Low | 1,319 | (15.7) | 492 | (37.3) | 827 | (62.7) | |
| **Number of Exercise Sessions per Week** | | | | | | | <0.0001 |
| More than four times | 5,347 | (63.7) | 2,321 | (43.4) | 3,026 | (56.6) | |

*(Continued)*

**Table 1.** (Continued)

| Variables | Total | | Dyslipidemia | | | | P-value |
|---|---|---|---|---|---|---|---|
| | | | Yes | | No | | |
| **Total** | **8,398** | **(100.0)** | **3,384** | **(40.3)** | **5,014** | **(59.7)** | |
| One-four times | 2,099 | (25.0) | 738 | (35.2) | 1,361 | (64.8) | |
| None | 952 | (11.3) | 325 | (34.1) | 627 | (65.9) | |
| **Parental History of Dyslipidemia** | | | | | | | 0.5204 |
| Yes | 8,011 | (95.4) | 3,222 | (40.2) | 4,789 | (59.8) | |
| No | 387 | (4.6) | 162 | (41.9) | 225 | (58.1) | |
| **Weight Change** | | | | | | | |
| Decrement | 5,408 | (64.4) | 2,149 | (39.7) | 3,259 | (60.3) | |
| Increment | 1,211 | (14.4) | 432 | (35.7) | 779 | (64.3) | |
| None | 1,777 | (21.2) | 803 | (45.2) | 974 | (54.8) | |
| **Year** | | | | | | | 0.0298 |
| 2013 | 1,607 | (19.1) | 634 | (39.5) | 973 | (60.5) | |
| 2014 | 1,398 | (16.6) | 548 | (39.2) | 850 | (60.8) | |
| 2015 | 1,591 | (18.9) | 663 | (41.7) | 928 | (58.3) | |
| 2016 | 1,837 | (21.9) | 787 | (42.8) | 1,050 | (57.2) | |
| 2019 | 1,965 | (23.4) | 752 | (38.3) | 1,213 | (61.7) | |

smokers and single smokers represented 3.0% and 37% of the total participants, respectively. Additionally, 38.4% of participants were ex-smokers and 21.6% were non-smokers.

Table 2 shows the association between smoking behaviors and prevalence of dyslipidemia among Korean men. Current and former users of either electronic or conventional cigarettes presented with a higher odd ratio of dyslipidemia compared to non-smokers. These results were statistically significant (ex-smoker: OR = 1.60, 95% CI = 1.41–1.83; single smoker: OR = 1.21, 95% CI = 1.06–1.38). Most importantly, those who smoked both conventional and electronic cigarettes were also at high risk of dyslipidemia (dual smoker: OR = 1.66, 95% CI = 1.26–2.20). Those who did not exercise had a higher risk of dyslipidemia than those who exercised regularly (>four times per week: OR = 0.80, 95% CI = 0.69–0.93; one-four times: OR = 0.79, 95% CI = 0.71–0.88). Higher self-reported health status was also related to a lower risk of dyslipidemia (high: OR = 0.71, 95% CI = 0.61–0.83; middle: OR = 0.85, 95% CI = 0.74–0.98). These results were all statistically significant. Finally, people who reported an increase in weight had a higher risk of developing dyslipidemia than those who maintained their weight (OR = 1.51, 95% CI = 1.34–1.70).

Table 3 shows the results of subgroup analyses between smoking behaviors and dyslipidemia, focusing on the household income level, self-reported health status, parental inheritance, number of exercise sessions per week. In every smoking group, people with lower middle household income level showed the highest odd ratios while at the same time being statistically significant. Such result may be due to the higher smoking prevalence among lower income individuals compared to those in upper middle or high groups [16]. Interestingly, those who answered 'high' on the health status report showed the highest odd ratios among every smoking group. Dual smokers who answered 'high' on the report were at 2.22 times higher prevalence compared to the non-smokers with the same answer. In fact, regardless of the health status, smoking has maleficent effects on individuals. Same conclusion can be deduced from the next variable. Except for the inherited dual smokers, all types of smoking resulted in higher odd ratios compared to the non-smoker with or without parental inheritance of dyslipidemia.

**Table 2. Factors associated with dyslipidemia.**

| Variables | Dyslipidemia | | P-value |
|---|---|---|---|
| | Male | | |
| | HR | 95% CI | |
| **Smoking behavior** | | | |
| Dual Smoker | 1.66* | (1.26–2.20) | 0.0004 |
| "Single" Smoker | 1.60* | (1.41–1.83) | < .0001 |
| Ex-Smoker | 1.21* | (1.06–1.38) | 0.0039 |
| Non-Smoker | 1.00 | | |
| **Age (years)** | | | |
| 20–39 | 0.85 | (0.71–1.01) | 0.0619 |
| 40–49 | 1.40 | (1.19–1.64) | < .0001 |
| 50–59 | 1.37 | (1.19–1.59) | < .0001 |
| ≥60 | 1.00 | | |
| **Educational level** | | | |
| Middle school or less | 1.22 | (1.04–1.44) | 0.0178 |
| High school | 1.14 | (1.01–1.28) | 0.038 |
| College or over | 1.00 | | |
| **Marital status** | | | |
| Married | 1.47* | (1.27–1.71) | < .0001 |
| Separated or divorced | 1.39 | (1.08–1.77) | 0.0091 |
| Unmarried | 1.00 | | |
| **Region** | | | |
| Urban area | 1.01 | (0.92–1.11) | 0.8383 |
| Rural area | 1.00 | | |
| **Household income level** | | | |
| Low | 1.04 | (0.88–1.23) | 0.6798 |
| Lower middle | 0.95 | (0.83–1.07) | 0.3823 |
| Upper middle | 0.95 | (0.85–1.07) | 0.4267 |
| High | 1.00 | | |
| **Drinking Status** | | | |
| Heavy drinking | 0.96 | (0.83–1.12) | 0.4434 |
| Moderate drinking | 0.96 | (0.87–1.07) | 0.6096 |
| Light drinking | 1.00 | | |
| **Occupational classification** | | | |
| White-collar | 1.14 | (0.98–1.33) | 0.0847 |
| Blue-collar | 0.96 | (0.84–1.11) | 0.5939 |
| Pink-collar | 1.09 | (0.91–1.29) | 0.3524 |
| None | 1.00 | | |
| **Self-Reported Health Status** | | | |
| High | 0.70* | (0.61–0.83) | < .0001 |
| Middle | 0.84* | (0.74–0.98) | 0.0229 |
| Low | 1.00 | | |
| **Stress Level** | | | |
| High | 1.12 | (0.96–1.30) | 0.1682 |
| Middle | 1.01 | (0.89–1.15) | 0.8856 |
| Low | 1.00 | | |
| **Number of Exercise Sessions per Week** | | | |
| More than four times | 0.80* | (0.69–0.93) | < .0001 |

(*Continued*)

**Table 2.** (Continued)

| Variables | Dyslipidemia | | P-value |
| --- | --- | --- | --- |
| | Male | | |
| | HR | 95% CI | |
| One-Four times | 0.79* | (0.71–0.88) | 0.0034 |
| None | 1.00 | | |
| **Parental Inheritance** | | | |
| Inheritance | 1.20 | (0.97–1.49) | 0.0928 |
| None | 1.00 | | |
| **Weight Change** | | | |
| Decrement | 0.88 | (0.77–1.01) | 0.0623 |
| Increment | 1.51* | (1.34–1.70) | < .0001 |
| None | 1.00 | | |
| **Year** | | | |
| 2013 | 1.03 | (0.89–1.18) | 0.6956 |
| 2014 | 1.01 | (0.87–1.17) | 0.9222 |
| 2015 | 1.11 | (0.97–1.28) | 0.1299 |
| 2016 | 1.16* | (1.01–1.32) | 0.0311 |
| 2019 | 1.00 | | |

Statistically significant was marked as

*. Abbreviations: CI, Confidence interval; OR, Odds ratio

Lastly, number of days engaged in muscular exercise was also associated with higher risks of dyslipidemia in those who dual smoked compared to non-smokers. Notably, this risks were also high for both those who have regular exercise cessions and those who exercise almost every day.

**Table 3. Subgroup analysis representing odds ratio for dyslipidemia stratified by smoking behavior.**

| Variables | Dual Smoker | | Single Smoker | | Ex-Smoker | | Non-Smoker |
| --- | --- | --- | --- | --- | --- | --- | --- |
| | OR | 95% CI | OR | 95% CI | OR | 95% CI | OR |
| **Household Income Level** | | | | | | | |
| Low | 2.20 | (0.73–6.66) | 1.51* | (1.02–2.23) | 1.29 | (0.88–1.89) | 1.00 |
| Lower middle | 1.82* | (1.02–3.25) | 1.88* | (1.43–2.48) | 1.36* | (1.03–1.79) | 1.00 |
| Upper middle | 1.37 | (0.84–2.24) | 1.61* | (1.26–2.05) | 1.20 | (0.93–1.54) | 1.00 |
| High | 1.75* | (1.10–2.80) | 1.49* | (1.20–1.85) | 1.15 | (0.93–1.42) | 1.00 |
| **Self-reported health status** | | | | | | | |
| High | 2.22* | (1.33–3.69) | 1.74* | (1.40–2.16) | 1.41* | (1.15–1.74) | 1.00 |
| Middle | 1.48* | (1.01–2.17) | 1.55* | (1.30–1.86) | 1.12 | (0.93–1.35) | 1.00 |
| Low | 1.15 | (0.55–2.41) | 1.27 | (0.85–1.88) | 0.99 | (0.65–1.49) | 1.00 |
| **Parental Inheritance** | | | | | | | |
| Inheritance | 1.56 | (0.51–4.78) | 2.44* | (1.61–3.69) | 1.98* | (1.33–2.94) | 1.00 |
| None | 2.78* | (2.15–3.59) | 2.52* | (2.29–2.77) | 1.72* | (1.57–1.89) | 1.00 |
| **Number of Exercise Sessions per Week** | | | | | | | |
| More than four times | 1.53* | (1.08–2.16) | 1.55* | (1.32–1.82) | 1.17 | (1.00–1.38) | 1.00 |
| One-four times | 1.98* | (1.13–3.49) | 1.64* | (1.25–2.14) | 1.16 | (0.89–1.51) | 1.00 |
| None | 1.81 | (0.70–4.66) | 1.86* | (1.20–2.87) | 1.42 | (0.97–2.07) | 1.00 |

Statistically significant was marked as

*. Abbreviations: CI, Confidence interval; OR, Odds ratio

## 4. Discussion

Due to the lack of authoritative information about different smoking behaviors, smokers often neglect the potential adverse effects of dual smoking in the belief that both single and dual smoking achieve the same results. Thus, in this study, we detailed the connections between dual smoking and dyslipidemia, using demographic, socioeconomic, and health-related variables gained from the 2013, 2014, 2015, 2016, 2019 KNHANES data. There was a positive association with dual smoking and a statistically significant increased risk of dyslipidemia; people who smoked both conventional and electronic cigarettes had a higher risk of developing dyslipidemia compared with non-smokers.

As many previous studies have proven the negative health impacts of conventional cigarettes, the South Korean government has implemented multiple anti-smoking regulations to reduce tobacco addiction [17]. As a result, many Korean men attempted to quit smoking, and often did so by using electronic cigarettes as substitutes for conventional cigarettes. Yet, many smokers ended up smoking both types of cigarettes rather than quitting. Our study illustrates the high potential harmful contribution of dual smoking to dyslipidemia compared to non-smokers and this presents a clear and advantageous course of action for people who are trying to quit smoking, suggesting that it is best to quit smoking of any kind, and choose means of smoking cessation other than electronic cigarettes.

The result from self-reported health status substantiates the notion that dual smoking has a negative association with developing dyslipidemia. Many studies show that the main intrinsic motivation for smoking cessation is health concerns [18]. Considering the large amount of information regarding the negative health impacts of conventional and electronic cigarette smoking, it is understandable that those who smoked checked "low" on the self-reported health status surveys. As a result, people who reported lower health status represent a larger proportion of dyslipidemia compared to those who reported "high".

In addition, regular exercise reflects a heightened possibility that someone is a non- or ex-smoker. According to a previous study, smoking is detrimental to physical fitness even among relatively young and fit individuals [19]. Smokers have lower physical endurance than non-smokers, even when the differences in average exercise levels between smokers and non-smokers are considered. This demonstrates the strong incentive for people who regularly workout to never smoke at all, or to stop smoking to improve physical fitness [19]. Similarly, those who gained weight within the previous year demonstrate the same point. People who regularly exercise have a higher chance of maintaining their weight compared to those who do not. As a result, those who exercise more than once a week comprise a smaller percentage of the total number of people with dyslipidemia than those who never workout.

Our study has several limitations. First, the data in this study are based on self-reported measures, and health status measurements might be subject to recall bias. Therefore, caution should be taken when interpreting these results. Second, due to this study's cross-sectional design, cause and effect, as well as the direction of the relationships observed, could not be determined. Third, we excluded Korean women from the study due to reporting bias. According to a study on the reporting bias of Korean women, the prevalence of self-reported smoking was 47.8% in Korean men and 6.6% in Korean women; however, the prevalence of smoking as assessed by urinary cotinine levels was 52.2% in men and 14.5% in women [13].

Despite these limitations, our study does possess several strengths. The KNHANES is conducted by a national institution based on random cluster sampling, and therefore, the data gained from it is statistically reliable and representative in comparison to surveys performed by private institutions. Moreover, as this study was conducted for over six years using variables developed by the KNHANES, the representativeness of the sample was improved upon.

Furthermore, KNHANES data is derived from health interviews, which includes both physical examinations and nutrition surveys, that form a reliable base for the creation of health-related policies and programs [20]. Therefore, this study can be a factor for motivating smokers to quit smoking entirely rather than transitioning to electronic cigarettes.

## 5. Conclusions

The general public's awareness of the negative health impact of dyslipidemia has increased tremendously due to the myriad of studies previously conducted [21]. Likewise, many prior studies also have proven that smoking conventional cigarettes results in a multitude of deleterious coronary diseases, including dyslipidemia [22]. Therefore, as electronic cigarettes were introduced, many smokers utilized them with the firm belief that electronic cigarettes are a healthier alternative to conventional cigarettes [23]. However, in reality, most of the smokers end up smoking both types of cigarettes and only few succeed in the transition. Since there is insufficient information regarding the health impacts of dual smoking; our research identified the relationship between dual smoking and the prevalence of dyslipidemia. We discovered that compared to non-smokers people who smoke both conventional and electronic cigarettes are more likely to develop dyslipidemia. Considering the asymptomatic nature of dyslipidemia, a full understanding of the associated risk factors could be the best solution to identify preventive measures this condition. The results of this study could be used to inform dual smokers to quit smoking entirely, or at least either kind of cigarette, and conventional cigarette single smokers to never attempt utilizing electronic cigarettes as smoking cessation tools, through its detailing and association with smoking behaviors and total cholesterol, HDL cholesterol, LDL cholesterol, and triglyceride levels.

## Author Contributions

**Conceptualization:** Wonseok Jeong.

**Formal analysis:** Wonseok Jeong.

**Methodology:** Wonseok Jeong.

**Supervision:** Wonseok Jeong.

**Writing – original draft:** Wonseok Jeong.

**Writing – review & editing:** Wonseok Jeong.

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
