## [Decision Letter · Decision Letter 0]

18 Apr 2022

PONE-D-22-06180Association Between Smoking Behaviors and Dyslipidemia in South Korean AdultsPLOS ONE

Dear Dr. jeong,

Thank you for submitting your manuscript to PLOS ONE. After careful consideration, we feel that it has merit but does not fully meet PLOS ONE’s publication criteria as it currently stands. Therefore, we invite you to submit a revised version of the manuscript that addresses the points raised during the review process.

We look forward to receiving your revised manuscript.

Kind regards,

Negar Rezaei, M.D., Ph.D.,

Academic Editor

PLOS ONE

Journal Requirements:

a) Did participants provide their written or verbal informed consent to participate in this study?

"No"

"No"

Reviewers' comments:

Reviewer's Responses to Questions

**Comments to the Author**

1. Is the manuscript technically sound, and do the data support the conclusions?

Reviewer #1: Partly

Reviewer #2: Partly

Reviewer #3: Partly

2. Has the statistical analysis been performed appropriately and rigorously? 

Reviewer #1: Yes

Reviewer #2: Yes

Reviewer #3: No

3. Have the authors made all data underlying the findings in their manuscript fully available?

Reviewer #1: Yes

Reviewer #2: Yes

Reviewer #3: No

4. Is the manuscript presented in an intelligible fashion and written in standard English?

Reviewer #1: Yes

Reviewer #2: No

Reviewer #3: Yes

5. Review Comments to the Author

Reviewer #1: Dear Editor,

This article aims to investigate the association between smoking behaviors including dual smoking (cigarette smoking + e-cigarette), single smoking, and ex-smoking with dyslipidemia using multiple logistic regression. The paper is well-written and well-crafted and the findings are of interest; however, there are some major concerns raised:

1. It would be suggestive to be more attentive about the interpretation of the study findings:

a) The authors designed a cross-sectional study to investigate the association between e-cigarettes and dyslipidemia but they modeled dual smoking, which does not correctly reflect the main effect of the e-cigarettes, alone. Therefore, it could be suggested that to discuss and interpret the findings with caution. I highly recommend that the authors go through the manuscript and revise the sections.

b) The ORs for dual smoking and single smoking were 1.66 (95% CI = 1.26–2.20) and 1.60 (95% CI =1.41–1.83), respectively. Kindly note that the confidence intervals are crossing and thus there is no statistically significant difference between the ORs of dual and single smoking. Again, I would recommend to be prudent regarding the effect of e-cigarettes on dyslipidemia.

Minor comments:

m1) Abstract: The conclusions section of the abstract does not inevitably follow the ideas presented in the previous sections of the abstract. Please make sure that all parts of the abstract follow one storyline and are consistent with each other.

m2) The introduction is too lengthy and hard-to-follow. Please make it to-the-point.

m3) Results: In some parts, the author explained ORs with with words such as "prevalence" as in "Current and former smokers of either electronic or conventional cigarettes presented with a higher prevalence of dyslipidemia compared to non-smokers. These results were statistically significant (ex-smoker: OR = 1.60, 95% CI =1.41–1.83; single smoker: OR = 1.21, 95% CI = 1.06–1.38)." Please go through the manuscript and revise such sentences.

Reviewer #2: This study investigates the association between smoking behavior and dyslipidemia using a database in South Korea. The author states that compared to non-smokers and ex-smokers, both single smokers and dual smokers had higher risk of being diagnosed with dyslipidemia, while dual smokers having the highest risk.

This is a well-written study. I think this study fits within the scope of PLOS ONE, and it seems to be an interesting paper for the research community in general. Clearly, the large sample size and national institution based random cluster sampling represent strengths of this investigation. Yet, I have some concerns that I would like to address.

Major:

1. [Tables] It would be a good idea to mark the data that are statistically significant in Table 2. Also, it would be better to add p-values in the table.

2. Conventional cigarettes are smoked while electronic cigarettes are vaped. I disagree with the phrase "dual smokers," "single smokers," and "electronic cigarettes smokers. "I advise the author to revise the phrase.

3. The introduction addresses the conventional smoking rate of Korean men, but that of electronic cigarettes is not mentioned. Adding the current smoking rate of electronic cigarettes among Koreans would strengthens the point that a lot of smokers are utilizing electronic cigarettes for smoking cessation method.

4. In US, each state has different policies. Regarding cigarettes, for example, San Franciso's Board of Supervisors voted unanimously to ban the sale and distribution of e-cigarettes in the city. I advise the author to add a current policy of Korean government regarding electronic cigarettes in there is any.

5. It would be better if subgroup analysis of few main variables are added. Adding the subgroup analysis of few key variables may help further assessing the relationship between smoking behaviors and dyslipidemia.

Reviewer #3: This article aims to investigate the effect of cigarette smoking on dyslipidemia in South Korean adults while including dual smoking as a critical indicator. The subject is novel and interesting, but the study design is unsuitable for the raised question. The authors must correct and explain these points:

Title

1. Please consider revising the manuscript title. The title needs to make the main question of the study crystal clear.

Abstract

1. Objective: "However, due to the high addictiveness of cigarettes, most individuals end up utilizing both types of cigarettes." The objective of the manuscript, as presented, does not inevitably follow this sentence. I think the rationale of the study objective is missing here.

2. Methods: Please clarify whether the Korean National Health and Nutrition Examination Survey is representative of the Korean population.

3. Methods: Like dyslipidemia, the authors need to present their definition of "smoking status".

4. The results section of the abstract does not correctly reflect the main manuscript's results section. The abstract must be self-explanatory and reflect the most eye-catching researchers' findings. Please revise the abstract and check the representativeness of the core ideas presented in the manuscript.

5. Conclusion: The conclusions section of the abstract does not inevitably follow the ideas presented in the previous sections of the abstract. Please make sure that all parts of the abstract follow one storyline and are consistent with each other.

Introduction

1. Consistent with the manuscript objective, the main focus of the study is on the shifting trends in smoking behaviors. Instead, a significant focus of the introduction has been on dyslipidemia. Please revise the introduction's storyline.

2. The current prevalence of all types of smoking is missing in the introduction. The authors need first to present the current situation of tobacco smoking in South Korea.

3. The magnitude of the problem investigated needs to be clarified in the introduction. Instead of subjective quantifiers such as "many", please consider presenting more the absolute values of the epidemiologic measures. For example What proportion of ever/ex/current cigarette smokers switch to e-cigarettes? What proportion quit cigarette smoking successfully after switching to e-cigarettes?

4. Is there any evidence of the deleterious effects of e-cigarettes in any combinations with other tobacco products? This needs to be addressed in the introduction.

Methods

1. The methods section lacks cohesion and storyline. Please revise.

2. The inclusion criteria are somewhat unclear. Were men currently smoking at each sprint of the survey included in the study?

3. The merger processes of 2013, 2014, 2015, 2016, and 2019 Korean National Health and Nutrition Examination Survey (KNHANES) data is unclear. Was this a pooled analysis? How did the authors handle duplicates? Were the sampling methods of each survey consistent with others?

4. As the study uses secondary data from KNHANES, more details on KNHANES study protocol need to be provided. Is KNHANES similar to the STEPwise approach to non-communicable disease risk factor surveillance (STEPS) proposed by World Health Organization? The methods section needs to be drafted to empower potential study duplication in the future.

5. The authors need to provide sufficient citations for all the data sources, definitions, etc., presented in the methods.

6. The basis for the occupations classifications needs to be cited and presented in methods.

7. The statistical analysis is insufficiently described. Ensure that all the results presented in the manuscript are derived from the methods presented in this section.

Results and Tables

1. Tables 1: Single smokers need to be defined. How many just smoked e-cigarettes/conventional cigarettes?

2. Table 2: The roles of e-cigarettes and conventional cigarettes need to be analyzed separately.

Discussion

1. Please initiate the discussion section with the most eye-catching findings of your study.

2. It is necessary to mention the authors' understanding of the article findings well.

3. Please review the discussion, check its storyline, and improve its coherency. It is not easy to follow in its current form.

6. PLOS authors have the option to publish the peer review history of their article (what does this mean?). If published, this will include your full peer review and any attached files.

Reviewer #1: **Yes: **Seyyed-Hadi Ghamari

Reviewer #2: No

Reviewer #3: **Yes: **Mohsen Abbasi-Kangevari

---

## [Author Response · Author response to Decision Letter 0]

17 May 2022

I was pleased to have the opportunity to revise my paper. In revising the paper, I have carefully considered your comments and suggestions. As instructed, I have attempted to explain the changes made in reaction to all of the reviewers’ comments. The reviewers’ comments were very helpful overall, and I appreciate the constructive feedback on my original submission. After addressing the issues raised, I feel the quality of the paper has greatly improved and I hope you agree. My response to each comment is as follows, and I attach a revision note with the highlighted, revised sections of the manuscript. Again, thank you for the valuable and helpful comments.

Response to Reviewer #1’s comments

The paper is well-written and well-crafted and the findings are of interest; 

Thank you for your great efforts in reviewing my manuscript. After the review, I totally revised the manuscript. 

Major comments: The authors designed a cross-sectional study to investigate the association between e-cigarettes and dyslipidemia but they modeled dual smoking, which does not correctly reflect the main effect of the e-cigarettes, alone. Therefore, it could be suggested that to discuss and interpret the findings with caution. I highly recommend that the authors go through the manuscript and revise the sections.

Response: Thank you for your comment. I totally understand your comment. I intended a study to investigate the association between dual smoking and dyslipidemia. Therefore, I have corrected phrases which indicated the association between e-cigarettes and dyslipidemia by changing “relationship between smoking behaviors and the prevalence of dyslipidemia, while including both conventional and electronic cigarettes as core indicators to “relationship between dual smoking and the prevalence of dyslipidemia” (revised manuscript, line 255~257). Also, in order to minimize the confusion of the purpose of the study, I have changed the title of the manuscript from “Association Between Smoking Behaviors and Dyslipidemia in South Korean Adults” into “Association Between Dual Smoking and Dyslipidemia in South Korean Adults”

Revised manuscript, line 255~257: Since there is insufficient information regarding the health impacts of dual smoking; our research identified the relationship between dual smoking and the prevalence of dyslipidemia.

Major comments: The ORs for dual smoking and single smoking were 1.66 (95% CI = 1.26–2.20) and 1.60 (95% CI =1.41–1.83), respectively. Kindly note that the confidence intervals are crossing and thus there is no statistically significant difference between the ORs of dual and single smoking. Again, I would recommend to be prudent regarding the effect of e-cigarettes on dyslipidemia.

Response: Thank you for your comment. I totally agree with your comment. I tried to correct all the sentences which indicated the associations between dual and single smoking as follows: “Those who smoke both conventional and electronic cigarettes were at even higher risk of dyslipidemia (dual smoker: OR = 1.66, 95% CI = 1.26–2.20).” into “Most importantly, those who smoked both conventional and electronic cigarettes were also at high risk of dyslipidemia (dual smoker: OR = 1.66, 95% CI = 1.26–2.20). (revised manuscript, line 161~163)” and “We discovered that compared to non-smokers, ex-smokers, and single smokers, people who smoke both conventional and electronic cigarettes are the most likely to develop dyslipidemia” into “We discovered that compared to non-smokers, people who smoke both conventional and electronic cigarettes are more likely to develop dyslipidemia (revised manuscript, line 257~258)” Minimal changes of phrases were also made in order to minimize such confusions. Thank you for your meaningful comments.

Revised manuscript, line 30~32: Most importantly, those who smoke both conventional and electronic cigarettes were also, at high risk of dyslipidemia (dual smoker: OR = 1.66, 95% CI = 1.26–2.20).

Revised manuscript, line 161~163: Most importantly, those who smoked both conventional and electronic cigarettes were also at high risk of dyslipidemia (dual smoker: OR = 1.66, 95% CI = 1.26–2.20).

Revised manuscript, line 257~258: We discovered that compared to non-smokers people who smoke both conventional and electronic cigarettes are the more likely to develop dyslipidemia.

Minor comments: Abstract: The conclusions section of the abstract does not inevitably follow the ideas presented in the previous sections of the abstract. Please make sure that all parts of the abstract follow one storyline and are consistent with each other.

Response: Thank you for your comment. We totally agree with your comment. We clarified the abstract part by changing the sentence from objectives part of the abstract from “However, due to the high addictiveness of cigarettes, most individuals end up utilizing both types of cigarettes. Therefore, this study aimed to examine the effect of cigarette smoking on dyslipidemia in South Korean adults, while including dual smoking as a key indicator.” into “However, due to the lack of researches scrutinized the relationship between dual smoking and dyslipidemia, those who failed in cessation end up utilizing both types of cigarettes. Therefore, this study aimed to examine the effect of dual smoking on dyslipidemia in South Korean adults.” (revised manuscript, line 17~20) Also, I changed the conclusion of the abstract to be more consistent with the previous sections by changing from “This study suggests that electronic cigarettes may not be a safe alternative to conventional cigarettes, and cessation of both types is necessary for a healthy life.” into “This study suggests that along with conventional cigarettes, dual smoking negatively impacts dyslipidemia and smoking cessation of every type is necessary for a healthy life.” (revised manuscript, line 38~39) 

Revised manuscript, line 17~20: However, due to the lack of researches scrutinized the relationship between dual smoking and dyslipidemia, those who failed in cessation end up utilizing both types of cigarettes. Therefore, this study aimed to examine the effect of dual smoking on dyslipidemia in South Korean adults.

Revised manuscript, line 38~39: This study suggests that along with conventional cigarettes, dual smoking negatively impacts dyslipidemia and smoking cessation of every type is necessary for a healthy life.

Minor comments: The introduction is too lengthy and hard-to-follow. Please make it to-the-point.

Response: Thank you for your comment. I totally agree with your comment. I figured out that a significant focus is on dyslipidemia instead of smoking behaviors. Therefore, I erased unnecessarily sentences from introduction regarding dyslipidemia, such as “According to a 30-year follow up study, for each 10 mg/dL increase in total cholesterol among those under 50 years of age, overall death and cardiovascular death increase by 5% and 9%, respectively”, “Correspondingly, in South Korea, diseases related to dyslipidemia, such as cerebrovascular, cardiovascular, and other circulatory diseases, account for almost one fifth of the total deaths among adults; this ratio is gradually increasing. The prevalence rate of dyslipidemia among Korean adults increased from 32.4% in 1998 to 42.6% in 2011.” Also, term ‘dual smoking’ has been used instead of ‘smoking behavior’ in order to minimize the confusion. Thank you for your meaningful comment. 

Minor comments: Results: In some parts, the author explained ORs with words such as "prevalence" as in "Current and former smokers of either electronic or conventional cigarettes presented with a higher prevalence of dyslipidemia compared to non-smokers. These results were statistically significant (ex-smoker: OR = 1.60, 95% CI =1.41–1.83; single smoker: OR = 1.21, 95% CI = 1.06–1.38)." Please go through the manuscript and revise such sentences.

Response: Thank you for your comment. I totally agree with your comment. I changed such sentences by changing “Current and former users of either electronic or conventional cigarettes presented with a higher prevalence of dyslipidemia compared to non-smokers.” into “Current and former users of either electronic or conventional cigarettes presented with a higher odd ratio of dyslipidemia compared to non-smokers.” (revised manuscript, line 158~160) and “Current and former smokers of either electronic or conventional cigarettes presented with higher prevalence of dyslipidemia compared to non-smokers (ex-smoker: Odds ratio (OR) = 1.60, 95% confidence interval (CI) = 1.41–1.83; single smoker: OR = 1.21, 95% CI = 1.06–1.38).” into “Current and former users of either electronic or conventional cigarettes presented with a higher odd ratio of dyslipidemia compared to non-smokers. (ex-smoker: OR = 1.60, 95% CI = 1.41–1.83; single smoker: OR = 1.21, 95% CI = 1.06–1.38).” (revised manuscript, line 31~33) Thank you for your meaningful comment.

Revised manuscript, line 158~160: Current and former users of either electronic or conventional cigarettes presented with a higher odd ratio of dyslipidemia compared to non-smokers.

Revised manuscript, line 31~33: Current and former users of either electronic or conventional cigarettes presented with a higher odd ratio of dyslipidemia compared to non-smokers. (ex-smoker: OR = 1.60, 95% CI = 1.41–1.83; single smoker: OR = 1.21, 95% CI = 1.06–1.38).

Response to Reviewer #2’s comments

This is a well-written study. I think this study fits within the scope of PLOS ONE, and it seems to be an interesting paper for the research community in general. 

Thank you for your great efforts in reviewing my manuscript. After the review, I totally revised the manuscript. 

Major comments: [Tables] It would be a good idea to mark the data that are statistically significant in Table 2. Also, it would be better to add p-values in the table.

Response: Thank you for your comment. I added * next to the data that are statistically significant and p-values in Table 2. Also, I added * next to the data that are statistically significant in newly added Table 3. By marking clearly, this could help understanding our tables. We believe that this could improve our manuscript quality. 

Revised manuscript, line 187: [Table 2] Statistically Significant was marked *

Revised manuscript, line 187: [Table 2] P-value was added

Revised manuscript, line 188: [Table 3] Statistically Significant was marked *

Major comments: Conventional cigarettes are smoked while electronic cigarettes are vaped. I disagree with the phrase "dual smokers," "single smokers," and "electronic cigarettes smokers. " I advise the author to revise the phrase.

Response: Thank you for your comment. I tried to correct all the inappropriate expressions regarding vaping an electronic cigarettes by changing “smoking status” into “cigarette use status”, “smoked an electronic cigarette” into “vaped an electronic cigarette” and “smokers” into “users”. I apologize for the confusion and thank you for giving us a chance to revise these errors.

Major comments:

The introduction addresses the conventional smoking rate of Korean men, but that of electronic cigarettes is not mentioned. Adding the current smoking rate of electronic cigarettes among Koreans would strengthens the point that a lot of smokers are utilizing electronic cigarettes for smoking cessation method.

Response: Thank you for your comment. I totally agree with your comments. I added a sentence in the introduction which demonstrates the smoking rate of electronic cigarettes in 2010 among Korean smokers. Thank you again for providing a meaningful advise which would strengthen the manuscript. 

Revised manuscript, line 58~63: Hence, Korean smokers have undertaken various strategies to quit smoking. Since the introduction of electronic cigarettes in the country, many smokers have attempted to utilize them for smoking cessation; according to ITC surveys conducted in 2010, the electronic cigarettes trial use and current use of e-cigarettes were 11% and 7 %, respectively, among smokers in Korea.

Major comments:

 In US, each state has different policies. Regarding cigarettes, for example, San Franciso's Board of Supervisors voted unanimously to ban the sale and distribution of e-cigarettes in the city. I advise the author to add a current policy of Korean government regarding electronic cigarettes in there is any.

Response: Thank you for your comment. I totally agree with your comments. I, however, could not find any e-cigarettes specific policies in Korea yet. I believe e-cigarettes specific policies have to be made and therefore more researches related to e-cigarettes and dual smoking have to be conducted. I hope this study to be helpful in making a smoking policy of Korea in the future. Thank you again for providing me a chance to look up e-cigarettes regulation policies in Korea. 

Major comments:

It would be better if subgroup analysis of few main variables are added. Adding the subgroup analysis of few key variables may help further assessing the relationship between smoking behaviors and dyslipidemia.

Response: Thank you for your comment. I totally agree with your comment. I added subgroup analyses between smoking behaviors and dyslipidemia, focusing on the household income level, self-reported health status, parental inheritance, number of exercise sessions per week. Also, due to an additional table, I added a paragraph in the results section explaining the analysis. Thank you for providing me an opportunity to further develop the study. 

Revised manuscript, line 188: [Table 3] was added.

Revised manuscript, line 171~186: Table 3 shows the results of subgroup analyses between smoking behaviors and dyslipidemia, focusing on the household income level, self-reported health status, parental inheritance, number of exercise sessions per week. In every smoking group, people with lower middle household income level showed the highest odd ratios while at the same time being statistically significant. Such result may be due to the higher smoking prevalence among lower income individuals compared to those in upper middle or high groups. Interestingly, those who answered ‘high’ on the health status report showed the highest odd ratios among every smoking group. Dual smokers who answered ‘high’ on the report were at 2.22 times higher prevalence compared to the non-smokers with the same answer. In fact, regardless of the health status, smoking has maleficent effects on individuals. Same conclusion can be deduced from the next variable. Except for the inherited dual smokers, all types of smoking resulted in higher odd ratios compared to the non-smoker with or without parental inheritance of dyslipidemia. Lastly, number of days engaged in muscular exercise was also associated with higher risks of dyslipidemia in those who dual smoked compared to non-smokers. Notably, this risks were also high for both those who have regular exercise cessions and those who exercise almost every day.

Response to Reviewer #3’s comments

This article aims to investigate the effect of cigarette smoking on dyslipidemia in South Korean adults while including dual smoking as a critical indicator. The subject is novel and interesting, but the study design is unsuitable for the raised question. The authors must correct and explain these points:

Thank you for your great efforts in reviewing our manuscript. I appreciate your comments and fully understand your concerns for the problems you have mentioned.

Title:

Major comments: Please consider revising the manuscript title. The title needs to make the main question of the study crystal clear.

Response: Thank you for your comment. I totally understand what you mean. Since the main focus of the manuscript is on dual smoking, I have changed the title from “Association Between Smoking Behavior and Dyslipidemia in South Korean Adults” into “Association Between Dual Smoking and Dyslipidemia in South Korean Adults” Thank you for your meaningful advise. 

Abstract:

Major comments: Objective: "However, due to the high addictiveness of cigarettes, most individuals end up utilizing both types of cigarettes." The objective of the manuscript, as presented, does not inevitably follow this sentence. I think the rationale of the study objective is missing here.

Response: Thank you for your comment. We totally agree with your comment. We clarified the abstract part by changing from “However, due to the high addictiveness of cigarettes, most individuals end up utilizing both types of cigarettes. Therefore, this study aimed to examine the effect of cigarette smoking on dyslipidemia in South Korean adults, while including dual smoking as a key indicator.” into “However, due to the lack of researches scrutinized the relationship between dual smoking and dyslipidemia, a lot of those who failed in cessation end up utilizing both types of cigarettes. Therefore, this study aimed to examine the effect of dual smoking on dyslipidemia in South Korean adults.” (revised manuscript, line 17~20)

Revised manuscript, line 17~20: However, due to the lack of researches scrutinized the relationship between dual smoking and dyslipidemia, a lot of those who failed in cessation end up utilizing both types of cigarettes. Therefore, this study aimed to examine the effect of dual smoking on dyslipidemia in South Korean adults.

Major comments: Methods: Please clarify whether the Korean National Health and Nutrition Examination Survey is representative of the Korean population.

Response: Thank you for your comment. I totally agree with your comment. I added a sentence which clarify whether the KNHNES is representative of the Korean population as follows: 

Revised manuscript, line 22~24: The KNHANES is conducted by a national institution based on random cluster sampling, and therefore, the data gained from it is statistically reliable and representative of the Korean population in comparison to surveys performed by private institutions. 

Major comments: Methods: Like dyslipidemia, the authors need to present their definition of "smoking status".

Response: Thank you for your comment. I totally agree with your comment. I added a sentence which present the definition of “smoking status” just as I did it for dyslipidemia. “Cigarette use status was categorized as one of the four status: dual smoker, single smoker, non-smoker, and ex-smoker. (revised manuscript, line 25~26)” Thank you for your advise.

Revised manuscript, line 25~26: Cigarette use status was categorized as one of the four status: dual smoker, single smoker, non-smoker, and ex-smoker. 

Major comments: The results section of the abstract does not correctly reflect the main manuscript's results section. The abstract must be self-explanatory and reflect the most eye-catching researchers' findings. Please revise the abstract and check the representativeness of the core ideas presented in the manuscript.

Response: Thank you for your comment. I totally agree with your comment. I emphasized a sentence which shows the association between dual smoking and dyslipidemia by changing from “Those who smoke both conventional and electronic cigarettes were at highest risk of dyslipidemia (dual smoker: OR = 1.66, 95% CI = 1.26–2.20).” into “More importantly, those who smoke both conventional and electronic cigarettes were also, at high risk of dyslipidemia (dual smoker: OR = 1.66, 95% CI = 1.26–2.20). (revised manuscript, line 33~35)” Also, I added a sentence which reflects another result section of the manuscript. “Along with smoking status, those who did not exercise had a higher risk of dyslipidemia than those who exercised regularly and higher self-reported health status was also related to a lower risk of dyslipidemia.”

Revised manuscript, line 35~37: More importantly, those who smoke both conventional and electronic cigarettes were also, at high risk of dyslipidemia (dual smoker: OR = 1.66, 95% CI = 1.26–2.20).

Revised manuscript, line 35~37: Along with smoking status, those who did not exercise had a higher risk of dyslipidemia than those who exercised regularly and higher self-reported health status was also related to a lower risk of dyslipidemia.

Major comments: Conclusion: The conclusions section of the abstract does not inevitably follow the ideas presented in the previous sections of the abstract. Please make sure that all parts of the abstract follow one storyline and are consistent with each other.

Response: Thank you for your comment. I totally agree with your comment. I changed the conclusion of the abstract to be more consistent with the previous sections by changing from “This study suggests that electronic cigarettes may not be a safe alternative to conventional cigarettes, and cessation of both types is necessary for a healthy life.” into “This study suggests that along with conventional cigarettes, dual smoking negatively impacts dyslipidemia and smoking cessation of every type is necessary for a healthy life.” (revised manuscript, line 38~39) Thank to your previous advises, I believe that the revisions that I already made in different parts of the abstract would also help every part of the abstract to follow one storyline. 

Revised manuscript, line 38~39: This study suggests that along with conventional cigarettes, dual smoking negatively impacts dyslipidemia and smoking cessation of every type is necessary for a healthy life.

Introduction:

Major comments: Consistent with the manuscript objective, the main focus of the study is on the shifting trends in smoking behaviors. Instead, a significant focus of the introduction has been on dyslipidemia. Please revise the introduction's storyline.

Response: Thank you for your comment. I totally agree with your comment. I figured out that a significant focus is on dyslipidemia instead of smoking behaviors. Therefore, I erased unnecessarily sentences from introduction regarding dyslipidemia, such as “According to a 30-year follow up study, for each 10 mg/dL increase in total cholesterol among those under 50 years of age, overall death and cardiovascular death increase by 5% and 9%, respectively”, and “Correspondingly, in South Korea, diseases related to dyslipidemia, such as cerebrovascular, cardiovascular, and other circulatory diseases, account for almost one fifth of the total deaths among adults; this ratio is gradually increasing. The prevalence rate of dyslipidemia among Korean adults increased from 32.4% in 1998 to 42.6% in 2011.” Also, adding prevalence of e-cigarettes from your next advise would increase the focus on smoking behaviors. Thank you for your meaningful comment. 

Major comments: The current prevalence of all types of smoking is missing in the introduction. The authors need first to present the current situation of tobacco smoking in South Korea.

Response: Thank you for your comment. I totally agree with your comment. Since I already mentioned the estimation of conventional cigarette smoking rate of Koreans, additional information regarding the prevalence of e-cigarettes smoking among Koreans are added. Thank you again for your meaningful advise. 

Revised manuscript, line 59~63: Since the introduction of electronic cigarettes in the country, many smokers have attempted to utilize them for smoking cessation; according to ITC surveys conducted in 2010, the electronic cigarettes trial use, and current use of e-cigarettes were 11% and 7 %, respectively, among adult smokers in Korea.

Major comments: The magnitude of the problem investigated needs to be clarified in the introduction. Instead of subjective quantifiers such as "many", please consider presenting more the absolute values of the epidemiologic measures. For example What proportion of ever/ex/current cigarette smokers switch to e-cigarettes? What proportion quit cigarette smoking successfully after switching to e-cigarettes?

Response: Thank you for your comment. I totally agree with your comment. While adding the prevalence of different types of smoking, both the rate of smokers who tried electronic cigarettes and smokers who currently use the electronic cigarettes are added: 11% and 7% respectively. Thank you for your valuable advise. 

Revised manuscript, line 59~63: Since the introduction of electronic cigarettes in the country, many smokers have attempted to utilize them for smoking cessation; according to ITC surveys conducted in 2010, the electronic cigarettes trial use, and current use of e-cigarettes were 11% and 7 %, respectively, among adult smokers in Korea.

Major comments: Is there any evidence of the deleterious effects of e-cigarettes in any combinations with other tobacco products? This needs to be addressed in the introduction.

Response: Thank you for your comment. I totally agree with your comment. I figured out that evidence of the deleterious effects of solely on either conventional cigarettes or electronic cigarettes are prevalent, but such effects of combinations of several tobacco products are hard to find. Therefore, I find this manuscript’s finding of association between dual smoking and dyslipidemia to be more valuable in the sense. Thank you again for your meaningful advise. 

Methods:

Major comments: The methods section lacks cohesion and storyline. Please revise.

Response: Thank you for your comment. I totally understand your comment. In order to increase the cohesion of the methods, I added a paragraph which explains a basic feature of the study. Also, a flow chart of the study has been added to increase the understanding of the readers. Thank you for your meaningful advise. 

Revised manuscript, line 99~104: Of the 39,208 individuals who participated in the surveys, I first excluded those aged <19 years (n=31,022); those under 19 years were excluded since the minimum age for smoking in South Korea is 20 years. Among 31,022, 17,002 remained due to the missing covariates such as heavy drinking status, parental inheritance of dyslipidemia, and exercise sessions performed in a week. Lastly, females were excluded, and the final study sample size was 8,398.

Major comments: The inclusion criteria are somewhat unclear. Were men currently smoking at each sprint of the survey included in the study?

Response: Thank you for your comment. We totally understand your comment. I clarified a sentence which distinguished the current smokers and ex-smokers by changing from ““Do you smoke/vape a conventional/electronic cigarette?” into “In the past 30 days, have you smoked/vaped a conventional/ electronic cigarettes?”(revised manuscript: 111~113) We apologize for the confusion.

Revised manuscript, line 111~113: Those who reported “Yes” to either of the questions were asked the following follow-up question: “In the past 30 days, have you smoked/vaped a conventional/ electronic cigarettes?”

Major comments: The merger processes of 2013, 2014, 2015, 2016, and 2019 Korean National Health and Nutrition Examination Survey (KNHANES) data is unclear. Was this a pooled analysis? How did the authors handle duplicates? Were the sampling methods of each survey consistent with others?

Response: Thank you for your comment. I totally understand your comment. I added a sentence which would explain the question as follows. I apologize for the confusion.

Revised manuscript, line 93~98: Along with stratification of the geographic areas (16 provinces of South Korea), probability-clustered sampling methods were performed in two steps. First, the primary sampling units were formed by sex, 26 age groups, and 24 land and housing classes. Second, 20 families were randomly sampled in each primary sampling unit. Then, each member of the sampled family provided written informed consent. Thus, the data from KNHANES can be considered to represent the entire population of South Korea

Major comments: As the study uses secondary data from KNHANES, more details on KNHANES study protocol need to be provided. Is KNHANES similar to the STEP wise approach to non-communicable disease risk factor surveillance (STEPS) proposed by World Health Organization? The methods section needs to be drafted to empower potential study duplication in the future.

Response: Thank you for your comment. I totally understand your comment. I added a sentence which would explain the question as follows. I apologize for the confusion

Revised manuscript, line 125~127: From 2018, the ethics approval for the KNHANES was waived by the KCDC Institutional Review Board under the Bioethics & Safety Act and opened to the public. The study protocol has been priorly approved by the Institution’s ethics committee on research on humans. The board waived the need for informed consent, as the subjects’ records and information were anonymized and de-identified prior to analysis

Major comments: The authors need to provide sufficient citations for all the data sources, definitions, etc., presented in the methods.

Response: Thank you for your comment. I totally understand your comment. Indeed, citations are very important part of the manuscript. Thank you again for a significant reminder. 

Kang SH, Jung DJ, Choi EW, Cho KH, Park JW, Do JY. HbA1c Levels Are Associated with Chronic Kidney Disease in a Non-Diabetic Adult Population: A Nationwide Survey (KNHANES 2011–2013). PLOS ONE. 

Lee DW, Kim K, Baek J, Oh SS, Jang S-I, Park E-C. Association of habitual alcohol use on risk-taking behaviors while using a car: The Korean National Health and Nutrition Examination Survey 2009–2013. Accident Analysis & Prevention. 

Major comments: The basis for the occupations classifications needs to be cited and presented in methods

Response: Thank you for your comment. I totally understand your comment. I added a citation which states the Korean version of the Standard Classification of Occupations. Thank you again for a significant reminder.

Nam JY, Kim J, Cho KH, Choi Y, Choi J, Shin J, et al. Associations of sitting time and occupation with metabolic syndrome in South Korean adults: a cross-sectional study. BMC Public Health. 

Major comments: The statistical analysis is insufficiently described. Ensure that all the results presented in the manuscript are derived from the methods presented in this section.

Response: Thank you for your comment. I totally understand your comment. I added a part which one variable (stress level) of the statistical analysis is not explained. Thank you for the advice. 

Revised manuscript, line 124~129: The health-related characteristics included the participants’ frequency of heavy drinking, defined as more than seven glasses of alcohol, (less than once a month, once a week, and almost every day), the number of days in a week that exercise was performed (none, 1�2 days, 3�4 days, 5�6 days, and every day), self-reported health status (good, normal, and bad), self-reported stress level (high, middle, and low), parental history of dyslipidemia (yes, no), and weight changes.

Results and Tables:

Major comments: Tables 1: Single smokers need to be defined. How many just smoked e-cigarettes/conventional cigarettes?

Response: Thank you for your comment. I totally agree with your comments. Since the study was not structured to investigate on difference between conventional cigarettes and electronic cigarettes but on dual smoking and single smoking, single smokers are defined as smokers who is currently smoking either one of the smoking types. Thus I did not added numbers of e-cigarettes/conventional cigarettes only smokers. Yet, following your advise, general statistics of Korean smokers are added in order to help defining ‘single smokers’ indirectly. Thank you for your valuable advise. 

Revised manuscript, line 59~63: Since the introduction of electronic cigarettes in the country, many smokers have attempted to utilize them for smoking cessation; according to ITC surveys conducted in 2010, the electronic cigarettes trial use, and current use of e-cigarettes were 11% and 7 %, respectively, among adult smokers in Korea.

Revised manuscript,

Major comments: Table 2: The roles of e-cigarettes and conventional cigarettes need to be analyzed separately.

Response: Thank you for your comment. I totally agree with your comments. The main purpose of the study was not analyzing different effects of e-cigarettes and conventional cigarettes on dyslipidemia, but was analyzing that of dual smoking and single smoking. I, however, understand how such confusion has been made and in order to minimize such confusion, I have corrected phrases which indicated the association between e-cigarettes and dyslipidemia by changing “relationship between smoking behaviors and the prevalence of dyslipidemia, while including both conventional and electronic cigarettes as core indicators to “relationship between dual smoking and the prevalence of dyslipidemia” (revised manuscript, line 255~257). Also, I have changed the title of the manuscript from “Association Between Smoking Behaviors and Dyslipidemia in South Korean Adults” into “Association Between Dual Smoking and Dyslipidemia in South Korean Adults” in order to reduce the confusion. 

Revised manuscript, line 255~257: Since there is insufficient information regarding the health impacts of dual smoking; our research identified the relationship between dual smoking and the prevalence of dyslipidemia.

Discussion:

Major comments: Please initiate the discussion section with the most eye-catching findings of your study.

Response: Thank you for your comment. I totally understand your comment. By erasing unnecessarily sentence, I revised the beginning of the discussion as follows: “Due to the lack of authoritative information about different smoking behaviors, smokers often neglect the potential adverse effects of dual smoking in the belief that both single and dual smoking achieve the same results. This, in this study, I detailed the connections between dual smoking and dyslipidemia, using demographic, socioeconomic, and health-related variables gained from the 2013, 2014, 2015, 2016, 2019 KNHANES data. (revised manuscript, line 191~195).

Revised manuscript, line 191~195: Due to the lack of authoritative information about different smoking behaviors, smokers often neglect the potential adverse effects of dual smoking in the belief that both single and dual smoking achieve the same results. Thus, in this study, I detailed the connections between dual smoking and dyslipidemia, using demographic, socioeconomic, and health-related variables gained from the 2013, 2014, 2015, 2016, 2019 KNHANES data.

Major comments: It is necessary to mention the authors' understanding of the article findings well.

Response: Thank you for your comment. I totally agree with your comments. I added a sentence which summarize my own understanding of the article’s findings as follows: My study illustrates the high potential harmful contribution of dual smoking to dyslipidemia compared to non-smokers and this presents a clear and advantageous course of action for people who are trying to quit smoking, suggesting that it is best to quit smoking of any kind, and choose means of smoking cessation other than electronic cigarettes. (revised manuscript, line 203~208)

Revised manuscript, line 203~208: My study illustrates the high potential harmful contribution of dual smoking to dyslipidemia compared to non-smokers and this presents a clear and advantageous course of action for people who are trying to quit smoking, suggesting that it is best to quit smoking of any kind, and choose a means of smoking cessation other than electronic cigarettes.

Major comments: Please review the discussion, check its storyline, and improve its coherency. It is not easy to follow in its current form.

Response: Thank you for your comment. I totally agree with your comments. I believe the previous additions that I made from your advises would improve the general coherency of the discussion. Furthermore, I changed few ambiguous sentences from “The role of higher self-reported health status substantiates the notion that dual smoking has a negative association with developing dyslipidemia.” into “The result from self-reported health status substantiates the notion that dual smoking has a negative association with developing dyslipidemia. (revised manuscript 209~210)”, “In summation, people who reported lower health status represent a larger proportion of dyslipidemia compared to those who reported "high." into “As a result, people who reported lower health status represent a larger proportion of dyslipidemia compared to those who reported "high" (revised manuscript 214~215) and “Smokers have lower physical endurance than nonsmokers, if when the differences in average exercise levels between smokers and non-smokers are considered. This demonstrates the strong incentive for people who regularly workout to have never smoked, or to stop smoking to improve physical fitness” into “Smokers have lower physical endurance than nonsmokers, even when the differences in average exercise levels between smokers and non-smokers are considered. This demonstrates the strong incentive for people who regularly workout to never smoke at all, or to stop smoking to improve physical fitness (revised manuscript 218~222)”

 Revised manuscript, line 209~210: A The result from self-reported health status substantiates the notion that dual smoking has a negative association with developing dyslipidemia.

Revised manuscript, line 214~215: As a result, people who reported lower health status represent a larger proportion of dyslipidemia compared to those who reported "high"

Revised manuscript, line 218~222: Smokers have lower physical endurance than nonsmokers, even when the differences in average exercise levels between smokers and non-smokers are considered. This demonstrates the strong incentive for people who regularly workout to never smoke at all, or to stop smoking to improve physical fitness

---

## [Decision Letter · Decision Letter 1]

14 Jun 2022

Association Between Dual Smoking and Dyslipidemia in South Korean Adults

PONE-D-22-06180R1

Dear Dr. jeong,

We’re pleased to inform you that your manuscript has been judged scientifically suitable for publication and will be formally accepted for publication once it meets all outstanding technical requirements.

Kind regards,

Negar Rezaei, M.D., Ph.D.,

Academic Editor

PLOS ONE

Reviewers' comments:

Reviewer's Responses to Questions

**Comments to the Author**

1. If the authors have adequately addressed your comments raised in a previous round of review and you feel that this manuscript is now acceptable for publication, you may indicate that here to bypass the “Comments to the Author” section, enter your conflict of interest statement in the “Confidential to Editor” section, and submit your "Accept" recommendation.

Reviewer #1: All comments have been addressed

Reviewer #3: All comments have been addressed

2. Is the manuscript technically sound, and do the data support the conclusions?

Reviewer #1: Partly

Reviewer #3: Yes

3. Has the statistical analysis been performed appropriately and rigorously? 

Reviewer #1: Yes

Reviewer #3: Yes

4. Have the authors made all data underlying the findings in their manuscript fully available?

Reviewer #1: Yes

Reviewer #3: No

5. Is the manuscript presented in an intelligible fashion and written in standard English?

Reviewer #1: Yes

Reviewer #3: No

6. Review Comments to the Author

Reviewer #1: (No Response)

Reviewer #3: Association Between Dual Smoking and Dyslipidemia in South Korean Adults

I would like to thank the author for the extensive revision of the manuscript. I think the quality of the manuscript has dramatically improved. A few minor comments:

1. Please consider defining “dual smoking” upon the first appearance in both abstract and the manuscript.

2. Please avoide using personal pronoun i.e. “I, my” in scientific writing.

7. PLOS authors have the option to publish the peer review history of their article (what does this mean?). If published, this will include your full peer review and any attached files.

Reviewer #1: **Yes: **Seyyed-Hadi Ghamari

Reviewer #3: **Yes: **Mohsen Abbasi-Kangevari

---

## [Editor Report · Acceptance letter]

30 Jun 2022

PONE-D-22-06180R1 

Association Between Dual Smoking and Dyslipidemia in South Korean Adults 

Dear Dr. Jeong:

I'm pleased to inform you that your manuscript has been deemed suitable for publication in PLOS ONE. Congratulations! Your manuscript is now with our production department. 

Kind regards, 

on behalf of

Dr. Negar Rezaei 

Academic Editor

PLOS ONE